# Neutron Star Mergers: Probing the EoS of Hot, Dense Matter by Gravitational Waves

**Matthias Hanauske [1,2], Jan Steinheimer [2], Anton Motornenko [1,2], Volodymyr Vovchenko [1,2], Luke Bovard [1,2], Elias R. Most [1], L. Jens Papenfort [1], Stefan Schramm [2] and Horst Stöcker [1,2,3,*]**

[1]   Institute for Theoretical Physics, Goethe University, Max-von-Laue-Straße, 1, 60438 Frankfurt am Main, Germany; hanauske@fias.uni-frankfurt.de (M.H.); motornenko@fias.uni-frankfurt.de (A.M.); vovchenko@fias.uni-frankfurt.de (V.V.); bovard@th.physik.uni-frankfurt.de (L.B.); most@fias.uni-frankfurt.de (E.R.M.); papenfort@itp.uni-frankfurt.de (L.J.P.)

[2]   Frankfurt Institute for Advanced Studies, Ruth-Moufang-Straße, 1, 60438 Frankfurt am Main, Germany; steinheimer@fias.uni-frankfurt.de (J.S.); schramm@fias.uni-frankfurt.de (S.S.)

[3]   GSI Helmholtzzentrum für Schwerionenforschung GmbH, 64291 Darmstadt, Germany

*   Correspondence: stoecker@fias.uni-frankfurt.de; Tel.:+49-174-3281445

**Abstract:** Gravitational waves, electromagnetic radiation, and the emission of high energy particles probe the phase structure of the equation of state of dense matter produced at the crossroad of the closely related relativistic collisions of heavy ions and of binary neutron stars mergers. 3 + 1 dimensional special- and general relativistic hydrodynamic simulation studies reveal a unique window of opportunity to observe phase transitions in compressed baryon matter by laboratory based experiments and by astrophysical multimessenger observations. The astrophysical consequences of a hadron-quark phase transition in the interior of a compact star will be focused within this article. Especially with a future detection of the post-merger gravitational wave emission emanated from a binary neutron star merger event, it would be possible to explore the phase structure of quantum chromodynamics. The astrophysical observables of a hadron-quark phase transition in a single compact star system and binary hybrid star merger scenario will be summarized within this article. The FAIR facility at GSI Helmholtzzentrum allows one to study the universe in the laboratory, and several astrophysical signatures of the quark-gluon plasma have been found in relativistic collisions of heavy ions and will be explored in future experiments.

**Keywords:** heavy-ion collisions; binary neutron star mergers; QCD phase diagram; gravitational waves

---

## 1. Introduction

All four interactions can be described by gauge theories. Three of them have been found to be Yang–Mills theories and have been formulated within quantum electrodynamics (QED), weak interaction, and quantum chromodynamics (QCD), which describes strong nuclear interactions. Gravity itself is also expected to be a gauge theory, and there are different ways to formulate such a theory. The theory of neutron stars is in general a complicated interplay between all known forces, but if one restricts oneself to weakly magnetized neutron stars and equilibrated systems, only two forces dominate the system, namely the strongest (QCD) and the weakest force (gravity described with general relativity (GR)). Due to the different symmetry transformation groups of these two gauge theories, the covariant derivatives, and as a result the underlying field equations, are different, but the mathematical gauge theoretical structures are analogous. One main difference between the two gauge theories is the confinement of QCD, which prevents the internal color degrees of freedom of the quarks and gluonic fields from being observed from outside. As a result, the almost massless quarks

are bound by the strong nuclear interaction with massive hadronic particles. However, under extreme conditions a hadron-to-quark phase transition (HQPT) can occur. The elementary matter in the early phase of the universe, in relativistic collisions of heavy ions and in binary neutron star (BNS) mergers are examples of such extreme scenarios.

High-energy heavy-ion collision data are compatible with an HQPT, and during the last decades several observables have been presented which indicate a transition of confined hadronic particles to a deconfined phase of quarks and gluons. The existence of a chiral crossover transition in QCD, which can be linked to the eventual deconfinement of quarks and gluons at large temperatures, is confirmed by state-of-the-art lattice QCD simulations [1,2]. Several experiments in recent decades have presented results indicating that such a transition indeed occurs in high energy nuclear collisions [3–10]. As predicted by Csernai and coworkers [11], a strongly rotating quark-gluon plasma (QGP) is formed in non-central ultra-relativistic heavy-ion collisions and was detected by the STAR collaboration at the Relativistic Heavy Ion Collider (RHIC) at Brookhaven National Laboratory. This finding established the hottest, least viscous, and most vortical fluid ever to have been produced in a laboratory, and a remarkably different rotational behavior was observed, compared to similarly dense and hot hadronic matter [12].

In a BNS merger, the field contributions of QCD and GR are of the same order. QCD, however, is not solvable in the nonperturbative regime, and numerical solutions of QCD on a finite space-time lattice are still unable to describe neutron star matter or even finite nuclei or infinite nuclear matter. As a consequence, several effective theoretical models of the hadronic interaction have been proposed (see, e.g., [13,14]); however, these models are limited in their description in terms of moderating temperature and densities and are therefore believed to be not applicable in a BNS merger scenario. If one wants to extend these theories to higher densities and temperatures, it is believed that hadronic matter must undergo a phase transition to a deconfined state consisting of quarks and gluons, the QGP. Within the $T = 0$ limit, the transition from a hadronic model to a quark model in the interior of a compact star follow the Gibbs or Maxwell condition [15–17]. Combined finite-temperature models of the hadronic and quark interaction and experimental data from heavy-ion collisions show that a cross-over HQPT occurs for high temperature values at low baryonic chemical potentials $\mu$. For example, within the temperature-dependent Q$\chi$P model [18–21], an effective QCD-motivated hadronic chiral $SU(3)_L \times SU(3)_R$ model has been extended by a deconfined phase of quarks using a strong HQPT. The predicted QCD phase structure of this model is visualized in Figure 1 by showing the quark fraction $Y_q$ for symmetric and $\beta$-equilibrated matter in the ($T$-$\mu$)-plane (left picture). The right picture of Figure 1 demonstrates the smooth separate transitions of chiral symmetry restoration and deconfinement. However, models of this kind are still rare and the implementation of the predicted temperature-dependent equation of states (EOSs) in hybrid star merger simulations have only been recently performed [22,23].

The long-awaited event (GW170817) took place on 17 August 2017. The LIGO/Virgo collaboration detected a gravitational wave (GW) from a BNS merger, and $\sim$70 astronomical observatories and satellites found a correlated electromagnetic signal. Many numerical-relativity simulations of BNS mergers were investigated long before the detection of GW170817 (see [24–34] for more recent work). The emitted GWs, the interior structure of the generated hypermassive/supramassive neutron star (HMNS/SMNS), the impact of initial spin and mass ratio, the accurate measurement of the amount of ejected material from the merger, the synthetic light curves of the produced kilonova signal, the distribution of the abundances of heavy-elements, the impact of magnetic fields, and last but not least the temperature and density distributions of the produced remnant have been analyzed in detail. The EOSs used within most of the studies do not take care of an incorporation of an HQPT in the interior of the produced hot and dense remnant.

Multi-messenger GW astronomy has become possible nowadays, and the breakthrough of the detection of a GW from a BNS merger by the LIGO/Virgo collaboration [35] will be briefly summarized in Section 2. Section 3 discusses BNS merger scenarios in the context of the HQPT. The interior

temperature and density structure of a neutron star merger product and the evolution of the hot and dense matter inside the HMNS/SMNS will be analyzed and visualized in a $(T - \rho/\rho_0)$ QCD phase diagram. A summary, an outlook, and a discussion of the astrophysical consequences of our results will be presented in Section 4. If a strong HQPT occurs during the post-merger phase, it will be imprinted in the emitted GW-signal and might additionally contribute to the dynamically emitted outflow of mass.

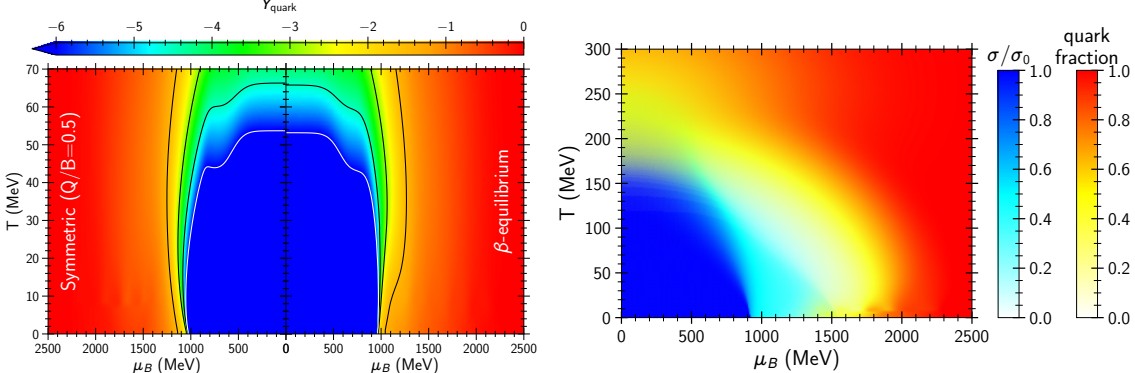

**Figure 1.** (**Left**) $(T\text{-}\mu)$ QCD phase diagram predicted by the Q$\chi$P model. The colormap indicates the logarithm of the quark fraction $Y_q$ for symmetric matter (left panel) and $\beta$-equilibrated neutron star matter (right panel). The contour lines of the quark fraction were taken at $Y_q = [0.0001]$ (white contour) and $Y_q = [0.001, 0.01, 0.1]$ (black contours); (**Right**) quark fraction and chiral condensate in the $(T\text{-}\mu)$ plane for isospin symmetric matter.

## 2. The New Era of Multi-Messenger Gravitational Wave Astronomy

The detection of a GW from a BNS merger by the LIGO/VIRGO collaboration (GW170817, [35]) marked the beginning of a new era in observational astrophysics. With the use of the observed tidal deformations of the two neutron stars from the late inspiral phase and other properties of GW170817, the EOS of dense matter could be severely constrained [36–44]. Unfortunately, the high density and temperature regime of the EOS within the post-merger phase was not observed in GW170817, but will possibly be detected in forthcoming merger events within the next observing run [45].

The neutron star merger scenario of GW170817 is in good agreement with the numerical simulations of BNS mergers performed in full general relativistic hydrodynamics. During the last several decades, a large number of numerical-relativity simulations of merging neutron star binaries have been performed, and the emitted GWs and the interior structure of the generated HMNS in the post-merger phase have been analyzed in detail. The extracted constraints on the mass of the total system of GW170817, in combination with the limitations on the EOS, which were estimated using the extracted tidal deformation of the two neutron stars right before merger [35], result in a neutron star merger scenario, which will be described in the following. As a result of the binary merger, a fast, differentially rotating compact object is produced, dubbed an HMNS. Matter in the interior of this object reaches densities of up to several times normal nuclear matter, and temperatures could reach $T \sim 50$–100 MeV. The numerical simulations show that, after the violent transient post-merger phase, the HMNS stabilizes after $\approx$10 ms, resulting in a quasi-stable configuration with a specific rotation profile [24]. The observation of the short gamma-ray burst GRB 170817A [46,47], which was detected with a time delay of $\simeq$1.7 s with respect to the merger time, indicates the collapse of the HMNS at a post-merger time $\simeq$1 s. In [37], the observations of GW170817 were used to constrain the maximum mass of neutron stars. The basic picture used within this approach is a scenario where the lifetime of the BNS merger product is assumed to be much longer than the timescale for reaching uniform rotation via magnetic braking (justified by the properties of the kilonova signal) and where the upper limit on the lifetime has been set by the observation of the GRB 170817A.

The merger is an extremely disruptive process, especially if the stars do not have the same mass or do not merge from quasi-circular orbits but through a dynamical capture [48]. Mass can be ejected either very rapidly—via tidal torques at the time of the dynamically merger or encounter—or more slowly—via winds that can be due to a number of different processes, which range from shock-heating to neutrino emission. This gravitationally unbound matter represents the perfect site for r-process nucleosynthesis and, if containing sufficient mass, can also lead to a bright electromagnetic signal, known as a "kilonova", as the material decays radioactively. In the follow-up observations of GW170817, a kilonova was observed providing the first definitive and undisputed confirmation of a kilonova and the formation of r-process elements from merging neutron stars [49,50].

During late post-merger times, the value of central rest-mass density can increase to several times of normal nuclear matter. However, for such high densities, the EOS is still poorly constrained, as the frequency spectrogram of the post-merger phase has not been detected in the GW170817 event [45]. By analyzing the power spectral density profile of the post-merger emission of a future event within the next observing run of the LIGO/VIRGO collaboration, the GW signal can set tight constraints on the high density regime of the EOS of elementary matter [51]. The modification of the EOS due to a potential influence of an HQPT and the impact of strange quark matter on the EOS, which is currently solely probed in relativistic heavy-ion collisions, might be imprinted in the post-merger phase of the emitted GW of a merging compact star binary. Hybrid star mergers represent therefore optimal astrophysical laboratories to investigate the phase structure of QCD and, in addition to the observations from heavy-ion collisions, will possibly provide a conclusive picture of the QCD phase structure at high density and temperature [52].

## 3. Numerical Relativity of Neutron Star Mergers and the Phase Diagram of QCD

In the following, we will use the theory of classical general relativity as the basis of our numerical neutron star merger simulations and neglect possible deviations coming from alternative formulations of the gravitational force. Einstein's theory of general relativity and the resulting general relativistic conservation laws for energy-momentum in connection with the rest-mass conservation are the theoretical groundings of neutron star binary mergers:

$$R_{\mu\nu} - \frac{1}{2}g_{\mu\nu}R = 8\pi\,T_{\mu\nu}\,,\ \ \nabla_\mu T^{\mu\nu} = 0\,,\ \ \nabla_\mu\left(\rho\,u^\mu\right) = 0. \tag{1}$$

$T_{\mu\nu}$ describes the energy-momentum tensor, $R_{\mu\nu}$ is the Ricci tensor, which contains first and second derivatives of the space-time metric $g_{\mu\nu}$, $\nabla_\mu$ is the covariant derivative, and $u^\mu$ is the four-velocity of the star's fluid.

The Einstein equation (first equation in Equation (1)) describes the way in which the space-time structure needs to bend (the left-hand side of the equation) if the energy-momentum is present (the right-hand side of the equation). These highly non-linear differential equations describe on the one hand how matter moves in a curved space-time and on the other hand formulates the way in which the amounts of energy-momentum curves the space-time structure. The ideal-fluid energy-momentum tensor $T_{\mu\nu} = (e + p)\,u_\mu u_\nu + p\,g_{\mu\nu}$ contains the energy and pressure densities of the nuclear and elementary particle physics contributions of the underlying neutron star matter, and $u^\mu = dx^\mu/d\tau$ describes the four-velocity of the star's fluid which is defined as the derivative of the coordinates $x^\mu = (t, x, y, z)$ by the proper time $\tau$.

To close the set of evolution equations, an EOS is needed. During the evolution of the BNS merger product, shocks will increase the temperature in certain regions of the HMNS/SMNS. Up to now, "hot", i.e., temperature-dependent EOSs are still rare; as a result, hybrid-thermal EOSs have been constructed by adding a thermal component to the "cold" EOSs available in various different elementary matter models. The pressure $p$ and the specific internal energy $\epsilon$ within these EOSs are composed of a "cold" part ($p_c$) and a "thermal" ideal-fluid component $p_{th}$ [53]:

$$p = p_c + p_{th} \quad \epsilon = \epsilon_c + \epsilon_{th}, \quad p_{th} = \frac{k_B}{m_N} \rho\, T \tag{2}$$

where $p$ and $\epsilon$ are the pressure and specific internal energy, respectively. The "thermal" part of the EOS is given by

$$p_{th} = \rho\, \epsilon_{th} \left( \Gamma_{th} - 1 \right) \qquad \epsilon_{th} = \epsilon - \epsilon_c \tag{3}$$

where $\Gamma_{th}$ is the thermal polytropic exponent.

The results of several exemplified BNS merger simulations will be presented in the following. The EOS used within the first framework is composed of a cold nuclear-physics part and a thermal ideal fluid component (for details, see [54,55]). The cold part has been modeled by a hybrid star matter model (ALF2-EOS [56]), where a phase transition to color-flavor-locked quark matter has been implemented. Within this model, the hadronic particles begin to deconfine to quark matter above a certain transition rest-mass density $\rho_{trans} = 3\rho_0$, where $\rho_0 := 2.705 \times 10^{14}$ g/cm$^3$ is the nuclear-matter rest-mass density. Assuming a moderate surface tension of the quark matter droplets, a phase transition is implemented by using a Gibbs construction (for details, see, e.g., [17]). As charge neutrality is only globally conserved within this construction, a mixed-matter phase exists in the rest-mass density range $3\rho_0 \leq \rho \leq 7.8\rho_0$.

For the used ALF2-EOS (an initial single star mass of the binary system $M_{In} = 1.35 M_\odot$ and a mass ratio of $q = 1$), the maximum density reached within the inspiral phase is below the onset of the HQPT, but soon after the merger the density reaches values above $\rho_{trans}$, forming a mixed phase inner region of deconfined quark matter. The detection of GWs from merging neutron star binaries can be used to determine the high-density regime of the EOS. The power spectral density profile of the post-merger emission is characterized by distinct frequency peaks and with the knowledge of the total mass of the system, the GW signal can set tight constraints on the EOS [51,55,57–60]. Although the ALF2-EOS has implemented an HQPT, the impact of the transition on the emitted GWs and star properties are small, as the softening of the EOS in the mixed phase region is weak. Additionally, the thermal effects are not implemented consistently within the ALF2-M135 run, as the thermal contributions to the EOS has been simply added using an thermal ideal fluid component.

In [61], the presumed appearance of a hadron–quark phase transition and the formation of regions of deconfined quark matter in the interior of a neutron star merger product have been addressed. A BNS merger simulation, which is based on the purely hadronic and temperature-dependent LS220-EOS (Lattimer–Swesty [62]) has been used to demonstrate that the evolution of the density and temperature profiles inside the inner region of the produced HMNS/SMNS advises an incorporation of an HQPT in the EOS of neutron star matter (see Figures 2 and 3). Figure 2 shows a comparison of the results of the LS220-M135 simulation with two typical simulations of a heavy-ion collision. The simulation is based on the LS220-EOS with a total gravitational mass of $2 \times 1.35\, M_\odot$ evolved without the assumption of $\pi$-symmetry (for details, see [24,63–65]).

For the heavy-ion collision simulations (see the gray and black curves in Figure 2), a numeric 3 + 1D relativistic ideal fluid dynamic simulation was performed by using the well-tested SHASTA algorithm [66]. The two heavy Au-nuclei were initialized as two counter-streaming Lorentz-contracted Wood–Saxon distributions with a central density of the well-known nuclear saturation density. After this initialization, the time evolution of the temperature and density of the whole colliding system can directly follow from the numerical algorithm, given the well-defined EOS [21]. The resulting evolution trajectories for the central cell of the heavy-ion collision is shown as lines in Figure 2. We compare here the compression and temperature for two different beam energies, i.e., two different collision velocities, as they are expected for experiments at the SIS18 accelerator at GSI.

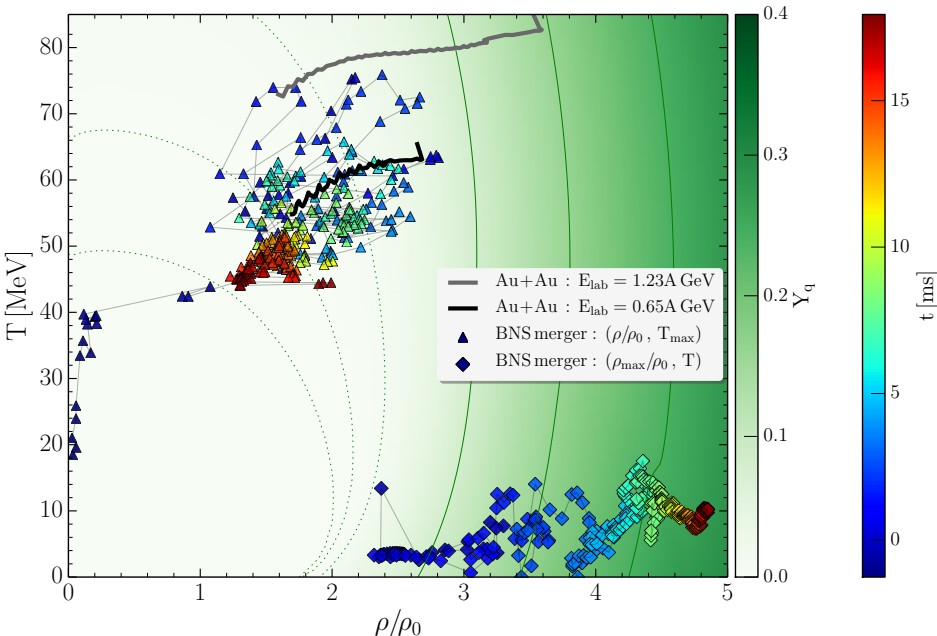

**Figure 2.** Evolution of the maximum value of the temperature (triangles) and rest-mass density (diamonds) at the equatorial plane in the interior of an HMNS using the simulation results of the `LS220-M135` run (for details, see [24,65]). The color coding of triangles/diamonds indicate the time of the simulation after the merger in milliseconds (see the colorbar at the right border of the figure). The gray and black curves show the trajectories of two heavy-ion collision simulations with energies $E_{\text{lab}} = 1.23$ A GeV and $E_{\text{lab}} = 0.65$ A GeV within the quark–hadron chiral parity-doublet model (Q$\chi$P model). The colorbar right next to the picture displays the quark fraction $Y_q$ of the corresponding hot and dense matter within the Q$\chi$P model. Contour lines of the quark fraction were taken at $Y_q = [0.0001, 0.001, 0.01]$ (green dotted lines) and $Y_q = [0.1, 0.2, 0.3]$ (green solid lines).

The density and temperature evolution of a BNS merger has similarities and differences with respect to a heavy-ion collision. The time evolution of the maximum value of the temperature in MeV (triangles) and rest-mass density in units of the nuclear matter density (diamonds) of the `LS220-M135` simulation are shown in Figure 2 in a ($T$-$\rho/\rho_0$) QCD phase diagram manner. The figure shows that the highly dense and hot neutron star matter of the remnant populates regions in the QCD phase diagram, where a non-negligible amount of deconfined quark matter is expected to be present (the green colorplot in the background of the picture indicates the quark fraction $Y_q$ within the Q$\chi$P model for symmetric matter).

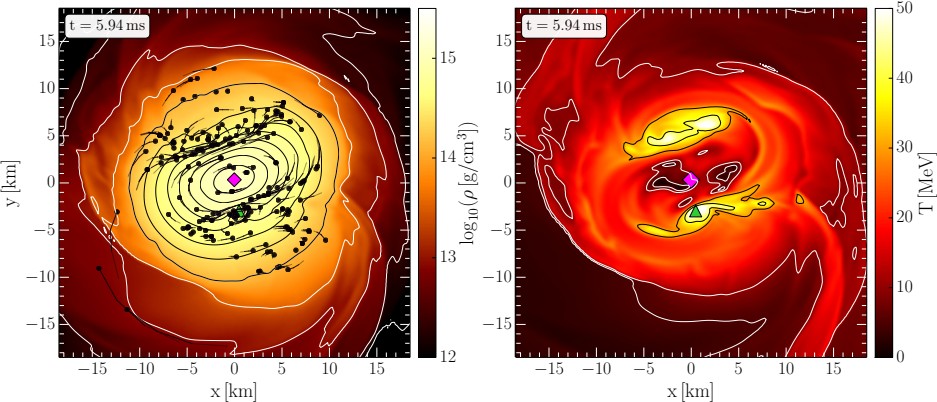

**Figure 3.** *Cont.*

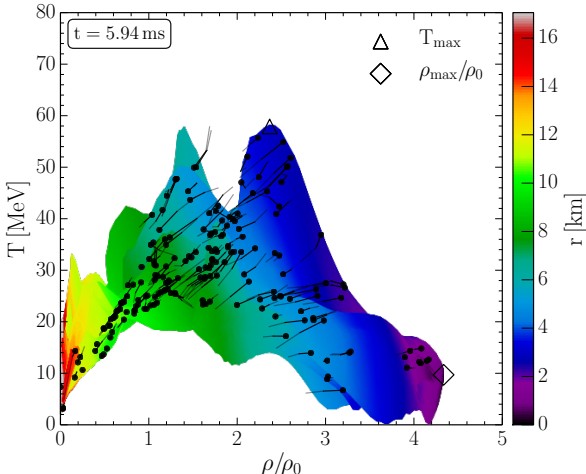

**Figure 3.** Upper pictures: Spatial distribution of the rest-mass density (**left**) and temperature (**right**) of the `LS220-M135` simulation at time $t = 5.94$ ms after merger. Lower picture: Density-temperature profiles inside the inner region of the HMNS in the style of a ($T$-$\rho/\rho_0$) QCD phase diagram plot. The color-coding indicate the radial position $r$ of the corresponding fluid element inside the HMNS. The open triangle/diamond marks the maximum value of the temperature/density. Additionally, several tracer particles that remain close to the equatorial plane are visualized with black dots. The final part of the tracer flowlines for the last $\Delta t = \simeq 0.19$ ms are shown, and the small black dots are used to indicate the tracer position at the time indicated in the frame. The initial parts of the trajectories have increasing transparency so as to highlight the final part of the trajectories.

In Figure 2, we neglected the structure of the density and temperature profiles of the HMNS and have only indicated the evolution of the maximum values of the temperature and density reached inside the neutron star merger product. The maximum values of the density (diamonds) and temperature (triangles) do not coincide spatially and the distributions of the rest-mass density and temperature profiles in the interior of the HMNS have a strongly spatial and time-dependent structure. A BNS merger simulation can be separated into different phases. Before the merger happens (inspiral phase), the temperature in the low-density regime, near to the region where the two NSs touch each other, increases rapidly, and at merger time the temperature hot spot of the newly born remnant reaches values up to $T \approx 75$ MeV. The maximum value of the density at $t = -0.5$ ms is below $2.5\,\rho_0$, and its spatial position is located in the center between the still separated two NSs. At merger time $t = 0$, where the emitted GW of the newly born remnant reaches its maximum value, the density maxima are almost at the center of the numerical grid and the high temperature regions are placed between them. The merger and the following violent, early post-merger phase $0 < t < 4$ ms are characterized by a pronounced density double-core structure and by hot temperature regions that are smeared out in areas between the double-core density maxima. After this post-merger phase, a new phase begins, and the high temperature regions transform to two temperature hot spots and move further out. The HMNS is stabilized by differential rotation, and the spatial structure of the rotation profile is deeply connected with the temperature structure (see the upper right picture in Figure 3) [24,61]. The density distribution at this post-merger time has a 'peanut' shape, but the highest value of the density $\rho$ is located in the center of the HMNS (see the left upper picture in Figure 3). The high temperature values ($T > 40$ MeV) are reached now in regions where the density is in a range of 1–2.5 $\rho_0$, while the maximum density values are always at moderate temperatures $T < 20$ MeV. The temperature hot spots have moved further out, and the interior of the HMNS, where the maximum of the density is located, has become denser. At later post-merger times, the temperature hot spots have smeared out to become a ring-like structure, the 'peanut' shape has been dissolved and the area populated in the ($T$-$\rho/\rho_0$) plane has been constricted to a small quasi-stable region (see left picture in Figure 4). The central region of the HMNS consists of highly dense matter ($\rho/\rho_0 \approx 5$) at moderate temperature

values $T \approx 10$ MeV, while the maximum of the temperature is reached at the top of the temperature ring-like structure at $r \approx 6$ km at moderate density values ($\rho/\rho_0 \approx 2$).

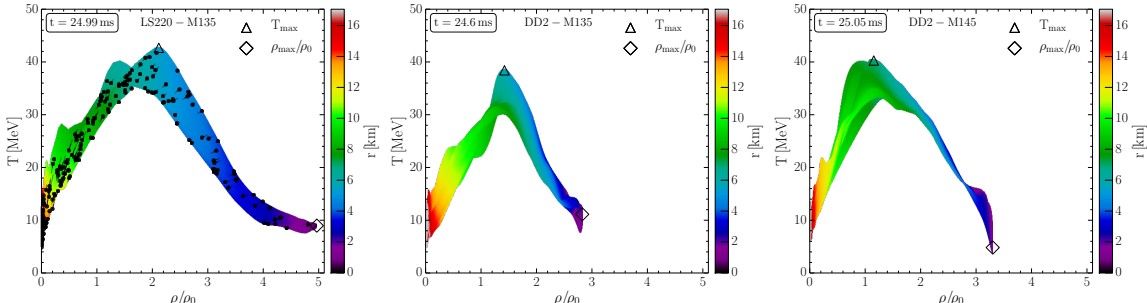

**Figure 4.** ($T$-$\rho/\rho_0$) density-temperature profiles of the `LS220-M135` (**left**), `DD2-M135` (**middle**), and `DD2-M140` (**right**) simulation at $t \approx 25$ ms.

Figure 4 compares the results of the late post-merger phase of the `LS220-M135` run with two simulations performed within the `DD2-EOS` (`DD2-M135` and `DD2-M140` simulation [65]). The figure shows the impact of the used EOS and total mass on the populate areas in the QCD phase diagram at a post-merger time of $t \approx 25$ ms. While the maximum value of the density reached within the `LS220-M135` run exceeds values $\rho/\rho_0 = 3.5$, the density of the HMNS within the `DD2-M135` simulation never reaches $\rho/\rho_0 = 3$. By increasing the total mass of the system, the density and temperature values reached during the post-merger phase increases (see the right picture in Figure 4).

The results of the `LS220-M135` and `DD2-M135/145` simulation have been based on temperature-dependent, but purely hadronic EOSs. The consequences of an appearance of the HQPT in the interior region of the neutron star merger product and its impact on the spectral properties of the emitted GWs has been recently discussed in [22,23,44,61,67]. Astrophysical signatures of the quark-gluon plasma in the interior of a compact star have been proposed within the last 20 years of research, and the results can be grouped in the following way (see Chapter 3.5 in [17]): Evidence of the HQPT is based on different mass and radius properties [68–70], rotational behaviors [71–74], twin star properties [68,75–77], and a future GW detection [17,22,23]. The effects of a strong HQPT have been investigated in the context of static [68,76,78] and uniformly rotating hybrid stars [71,72,74], and the results show that tremendous changes in the star properties might occur and show the existence of a third family of compact stars—the so-called "twin stars" [75].

In [22], a temperature-dependent chiral mean field (CMF) model [79] with a strong HQPT has been used for the first time in a BNS merger simulation. Due to the fact that the EOS does not yield gravitationally stable hybrid stars with deconfined quark matter cores, the effects of the HQPT can only be observed in a BNS merger scenario. In [22], it was shown that the phase transition, which happens during the post-merger phase, leads to a hot and dense quark core that, when it collapses to a black hole, produces a ringdown signal different from the hadronic one. The evolution of the temperature and density in the merger remnant is different from a purely hadronic model (see Figure 2 and Figure 3 in [22]), as the inner region of the hypermassive hybrid star forms a very hot and ultra-dense quark core before the collapse to a Kerr black hole. Similarities and differences are found when comparing the results of [22] with the evolution of the maximum values of the rest-mass density and temperature inside the HMNS presented within this article (see Figure 2 and Figure 3 in [22]). For the first $\approx 5$ ms after merger, both numerical simulations behave quite similarly. However, the formation of the hot and dense quark core for later post-merger times changes the evolution of the elementary matter in the phase diagram. In contrast to all of the hadronic models presented within this article, the temperature in the inner region of the HMNS increases in [22], resulting in a different evolution of the maximum temperature values.

In [23], the temperature-dependent, hadron–quark hybrid (DD2F-SF) model [80] has been used in a BNS merger simulation. In contrast to [79] stable hybrid stars, containing both hadrons and quarks

are realizable within the DD2F-SF EOS. In [23], it was shown that the dominant post-merger GW frequency $f_{\mathrm{peak}}$ exhibits a significant deviation from the empirical relation between $f_{\mathrm{peak}}$ and the tidal deformability $\Lambda$, if a strong first-order phase transition leads to the formation of a gravitationally stable extended quark matter core in the post-merger remnant. Such a shift of the dominant post-merger GW frequency might be revealed by future GW observations using second- and third-generation GW detectors.

Especially within an EOS that includes the possibility of twin star behavior, the astrophysical observables of an HQPT might be detectable by the future detection of neutron star merger events [44,61,67,81]. In [44], it was shown that using detections of GWs from BNS mergers with the same value for $\Lambda_1$ but different values of $\Lambda_2$ serves as a signal for the existence of a strong first-order phase transition in neutron star matter. Presupposing twin-star solutions and assuming that the hadronic part of the EOS is known up to a certain density, the global parameters of the HQPT are constrained tightly in order to explain the GW170817 event [81]. Binary hybrid star merger simulations that implement a strong HQPT, with the possibility of a twin-star behavior, are currently under construction, and the preliminary results show that the appearance of the HQPT in the interior region of the HMNS will change the spectral properties of the emitted GW, if the phase transition is strong enough. If the unstable twin-star region is reached during the post-merger phase, the $f_2$-frequency peak of the GW signal will change due to the speed increase of the differentially rotating HMNS, and large twin-star oscillations might occur [61,67].

## 4. Summary and Outlook

In this article, we show that the properties of elementary matter at high temperatures and densities can be studied in two different physical scenarios. High-energy heavy-ion collision experiments try to determine the phase structure of the isospin-symmetric QCD equation of state, and knowledge of the isospin-asymmetric QCD-EOS is needed in a general relativistic computer simulation of BNS mergers. It is therefore possible to study the properties of dense QCD for systems of different size, timescales, and chemical composition, which will eventually lead to an understanding of the properties of this elementary form of matter. Future experimental facilities, such as the FAIR (GSI) and NICA (Dubna), will explore the properties of compressed and hot elementary matter in heavy-ion collisions. With the use of these results, it will be possible to describe and understand future GW observations of BNS mergers of the LIGO/Virgo collaboration.

**Author Contributions:** Conceptualization, M.H. and H.S.; Methodology, M.H. and H.S.; Investigation, M.H., J.S., A.M., L.B., E.R.M., L.J.P., S.S. and V.V.; Writing—Original Draft Preparation, M.H.; Writing—Review & Editing, M.H., H.S., J.S., A.M. and S.S.; Funding Acquisition, H.S.

**Funding:** M.H. acknowledges support from Frankfurt Institute for Advanced Studies (FIAS) and the Institute for Theoretical Physics (ITP) at the Goethe University in Frankfurt. The support from European COST Actions "NewCompStar" (MP1304) and "PHAROS" (CA16214) and the LOEWE-Program of Helmholtz International Center for FAIR of the state of Hesse (Germany) is gratefully acknowledged.

**Acknowledgments:** We would like to thank Luciano Rezzolla. Without his profound knowledge and his comprehensive expertise in the field of numerical relativity and general relativistic hydrodynamics, this article and the simulations herein would not have been possible. Additionally, we would like to thank Glòria Montaña and Laura Tolos for valuable discussions.

**Conflicts of Interest:** The authors declare no conflict of interest.

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
