# Peer review of "Neutron Star Mergers: Probing the EoS of Hot, Dense Matter by Gravitational Waves"

_2571-712X, doi:10.3390/particles2010004_

Reviewer 1 Report

The manuscript "Neutron Star Mergers: Probing the EoS of Hot, Dense Matter by Gravitational Waves" summarizes the possibility to constrain the unknown Equation of State of matter at supranuclear densities with future multi-messenger observations of binary neutron star mergers. The authors present two numerical relativity simulations and discuss some of the features presented during the merger process. They relate the properties within the merger remnant with the QCD-phase diagram. They emphasize the strong synergy between relativistic collisions of heavy ions and binary neutron stars mergers.

Broad comments: 

The multimessenger observation of GW170817 marks a scientific breakthrough. This makes the topic of the paper very timely and potentially important for the readers and the broad scientific community. Unfortunately, the current form of the manuscript has a number of issues and problems. 

1.) As presented, the manuscripts seems not to be based on original work. It reuses a number of text passages and figures from J.Phys.Conf.Ser. 878 (2017) no.1, 012031 . 

For example Fig. 2 of the manuscript is identical with to Fig. 1 of their old publication. 

This seems not to be stated in the caption of the figure and seems also not been mentioned in the text. Similarly, the discussion about the numerical relativity methods seems only to be a slight modification of their 'originally' publication. Overall this repetition is visible throughout large parts of the paper. 

2.) After comparison with J.Phys.Conf.Ser. 878 (2017) no.1, 012031, my original interpretation of the manuscript was that the novelty of the work was the presentation of the CQG phase diagram in connection with binary neutron star simulations. However, such a phase diagram has been presented by the authors before, e.g., arxiv [1807.03684]. Thus, I do not see new scientific results in the current version of the manuscript. Similarly, simulations with the ALF2 and LS220 EOS seem to be published in a number of numerical relativity papers (if I am not mistaken).   

3.) Overall, the presentation and the reference selection seems to be extremely biased towards the work of the authors/Frankfurt research group. About half of all the references are self-citations. While it is understandable that several references refer to their own work [to ensure that code definitions etc. are correct], this selection bias is also presented at several other instances, e.g., 

the citation of  Phys.Rev.Lett. 120 (2018) no.26, 261103, but not Phys.Rev.Lett. 120 (2018) no.17, 172703; the citation of a number or works by Baiotti/Rezzolla/Takami for the study of the gravitational wave spectrum after merger, but no reference to the works of Bauswein et al., or Bernuzzi et al. ; 

Similarly it seem strange that Astrophys.J. 842 (2017) no.2, L10 is not cited, although it presented one of the first full numerical relativity works showing the imprint of the extreme density part of the equation of state -- very similar to ideas in the current manuscript. 

4.) It seems to me that the discussion about the (3+1)-split is not necessary for this paper. 

Overall, I suggest major revisions of the paper: 

The presentation should be changed to emphasize the use of the QCD-phase diagram and why this might be important for the field. Here it might be useful to compare the phase diagram of different simulations and Equations of state with each other. 

Furthermore, repetition to previous work should me minimized and references should be updated. 

Minor comments: 

abstract: 

"3+1 dimensional special- and general relativistic hydrodynamic simulation studies reveal a unique

window of opportunity to observe phase transitions in compressed baryon matter in statu nascendi by

multimessenger observations."

How do special relativity simulations help to reveal the nature of phase transition in connection with multimessenger observations? 

page 1: 

"One main difference between the two gauge theories is the confinement of QCD, which has the effect that the internal color degrees of freedom of the quarks and gluonic fields, cannot be observed from outside." 

It seems to me that the second comma in the sentence should be removed. 

page 3: 

"The detection of a gravitational wave from a binary neutron star merger by LIGO (GW170817, [43]) and the ensuing 1.7 seconds delayed gamma-ray burst [44] in addition with the electromagnetic counterparts of the associated kilonova [45] by numerous observatories around the world marked the beginning of a new era in observational astrophysics."

Although in the blind spot of Virgo, GW170817 still counts as a LIGO + Virgo detection. 

page 3:

"..., results in a neutron star merger scenario which is remarkably similar to the APR4-M135 simulation discussed in [56]."

I am unable to see the justification of this sentence, the chirp mass of the binary is not consistent with GW170817.

page 4: 

"The observation of the gamma-ray burst GRB 170817A [44], which was detected with a time delay of ' 1.7 s with respect to the merger time, indicates the collapse of the HMNS at a post-merger time ' 1 − 1.7 s."

This statement needs justification, either via references or additional text. Where does the lower bound of 1s comes from and what arguments do the authors use to justify that a black hole formed. 

page 5:

"In order to solve the evolution of a merging neutron star binary system numerically, Eq. (1) needs to be rewritten, because its structure is not well posed."

This sentence seems a bit sloppy. If the authors refer to the fact that they rewrite the Einstein Equations to obtain an initial value problem, then this should be stated clearly. 

page 5:

"As the ADM equations are still not ’well posed’ (for details see [62]), they need to be further transformed using a conformal traceless formulation."

While the authors discuss the (3+1)-split, they finally do not state which Equations are actually solved, this is rather unsatisfying. 

page 6: 

Is $M_{In}$ defined within the paper? 

page 7: 

Equation of state is spelled out, although it got abbreviated before.   

Author Response

We would like to thank the referee for his numerous and useful comments. The point-by-point response to the reviewer’s comments and the revised version of our manuscript (with and without highlighted corrections) has been attached below.

Reviewer 2 Report

Dear Editor, dear Authors

the paper by Hanauske et al. describes various aspects of hot and dense matter in the context of heavy-ion collision and neutron star mergers. The discussion is very broad and general and therefore helpful. I recommend publication of the manuscript after the authors took into account my comments below.

- On page 2 the authors write "In [7] it was estimated that the GW-frequency at the moment of collision in a neutron star merger (f_peak) is lower than in a hybrid or quark star merger and these estimates were recently confirmed by Bauswein et.al. [32]." I do not have access to Ref. [7]. However, if the authors refer to the Gw frequency at the moment of collision, then the scenario in [7] is different from the effect in [32], which discusses the impact on the post-merger frequency and not the frequency at the moment of the collision.

- Pages 3 and 4. The authors suggest that a short GRB was observed. This is not so clear. Considering the distance of the event the emission was much too subluminous to be a ordinary short GRB, although there is evidence that there was a relativistic outflow.

- The authors write "When comparing the produced light curves from different simulations with those observed, show that the simulation
results are significantly dimmer than those observed, which was due to a lower amount of ejected material and a lack of lanthanides [59,60]. This suggests that the dynamical ejecta are not the major
source of ejecta from a merger, but places a secondary role to other forms of secular ejecta, such as from neutrino driven winds or viscous ejecta from a disk." I think these are very strong statements, which are possibly wrong. First of all, the emission models that were used to infer the ejecta masses are rather simplistic and have significant uncertainties. Second, dynamical ejecta of the order of 0.02 Msun or even more has been found in some simulations for specific binary systems. Hence, it is not clear how much the dynamical ejecta contributed in this event. It may well be a major fraction considering also the uncertainties of all current simulations.

- Page 6: The authors discuss the spectral properties of the post-merger remnant. They refer to [64,67,68] for a discussion of the various frequencies of the post-merger signal. I would be appropriate to present here a more balanced list of references, as also other groups, e.g. Bauswein & Stergioulas, contributed substantially to the understanding of the frequency content of the spectrum (partially even before the mentioned references) and several issues are not fully settled yet.

- Page 6/Fig. 3. Diagrams like Fig. 3 are basically projections and dismiss information about the proton fraction. Does Y_q also depend on Ye? If so this should be mentioned and specified.

- Page 7: "The figure shows that the highly dense and hot neutron star matter of the remnant, populates regions in the QCD phase diagram where a non-neglectable amount of deconfined quark matter is expected to be present" It would be good to state here that this is an expectation for this chosen model.

Best regards

Author Response

We would like to thank the referee for his numerous and useful comments. The point-by-point response to the reviewer’s comments and the revised version of our manuscript (with and without highlighted corrections) has been attached below.

Round  2

Reviewer 1 Report

The authors have improved the manuscript.

One of my main issues, the missing novelty of the work, is still present. The authors write: 

"Yes, several BNS merger simulations of other groups have been performed using the ALF2 and LS220 EOS, however the visualization within the QCD phase diagram including tracer particles has never been addressed before. In order to increase the novelty of our scientific results, we have added an additional figure (Fig. 4) where we show a comparison between the results obtained within the LS220 and DD2 EOS and discuss the impact of the initial mass on the phase diagram structure."

A phase diagram has been presented in arxiv: 1807.03684. Furthermore, the authors do not discuss in detail the change of the phase diagram for the additional cases they have added. It is stated that with increasing initial mass, the central density increases, thus, different regions in the phase diagram are covered. Isn't this simply related to the fact that the setup is closer to black hole formation. 

I would recommend that the authors further extend their discussion about the phase diagram and their comparison of different configurations. Furthermore, to add new scientific results considered to previously published results: it seems plausible to discuss how potential measurements could be used to create such a phase diagram. Is such a description limited to numerical simulations or are we able to create a phase diagram from future observations? If so, how would this be done. 

Or asking differently: Can their method of constructing the phase diagram be used to constrain the equation of state and if so how would this be done explicitly. 

An additional minor comment is that the reference list seems not to be up do date. Several preprints have already been published.  

Author Response

Dear Editor,

Thank you for sending us the second report, who we thank for the numerous
and useful comments. Our response to your and the referee comments are addressed below
(our responses have been highlighted with "#########"). We hope that with the modifications
made the article will be considered acceptable for publication.

Best regards,
The authors

Second report from referee:

One of my main issues, the missing novelty of the work, is still present. The authors write:

"Yes, several BNS merger simulations of other groups have been performed using the ALF2 and LS220 EOS, however the visualization within the QCD phase diagram including tracer particles has never been addressed before. In order to increase the novelty of our scientific results, we have added an additional figure (Fig. 4) where we show a comparison between the results obtained within the LS220 and DD2 EOS and discuss the impact of the initial mass on the phase diagram structure."

A phase diagram has been presented in arxiv: 1807.03684. Furthermore, the authors do not discuss in detail the change of the phase diagram for the
additional cases they have added. It is stated that with increasing initial mass, the central density increases, thus, different regions in
the phase diagram are covered. Isn't this simply related to the fact that the setup is closer to black hole formation.

#########
The structure of the phase diagram presented in arxiv: 1807.03684 is quite different from the phase diagrams presented in Fig.2. We have added a brief discussion of the main differences in the revised version of our article (see page 9, upper part). We hope that this will also increase the novelty of the article.

The fact that the central density of the HMNS for late post merger times (see Fig.4) increases for larger values of the initial neutron star mass is mainly caused by a larger gravitational force which compresses the matter to higher density/pressure values. One might express this with "the setup is closer to black hole formation".
#########

I would recommend that the authors further extend their discussion about the phase diagram and their comparison of different configurations.
Furthermore, to add new scientific results considered to previously published results: it seems plausible to discuss how potential measurements could be used to create such a phase diagram. Is such a description limited to numerical simulations or are we able to create a phase diagram from future observations? If so, how would this be done.

Or asking differently: Can their method of constructing the phase diagram
be used to constrain the equation of state and if so how would this be
done explicitly.

#########
All of these questions are very interesting, and we are currently working on the issue: How a potential future measurement of a post merger GW could be used to construct the structure of the phase diagram of QCD.
#########

An additional minor comment is that the reference list seems not to be up
do date. Several preprints have already been published.

#########
We want to thank the referee for this comment. The reference list has been updated.
#########

Comment from the academic editor:

We request to address the technical and editorial changes suggested by the referees. The reference list should be made more balanced by including the work of other authors. This concerns both the simulations and the equation of state parts of the paper. A balanced reference list should contain roughly no more than about 25% self-citations, and the authors should try to adhere to this.

#########
We want to thank the editor for this comment. In the revised version, we have taken several of our articles out of the reference list.
#########

In addition, the topic of gravity as a gauge field theory with Ref. 1-7 seems to be out of the context. The rest of the paper uses GR as a classical theory and the results are obtained with the standard formulation of the GR. The quantum gravity is a broad field with many unsolved problems, therefore the brief excursion in this subject with reference only to own work and work of co-workers appears not to be adequate. The authors need either to drop the topic of quantum gravity or at least include several citations to the recent work and reviews on the subject by other authors or groups.

#########
In order to take this comment into account and concurrently decrease the number of self-citations, we have taken out the references [1-8] in the revised version.
#########

... However, the authors should properly refer to already published material, for example by adding "Figure taken from [insert reference.]"

#########
None of the figures presented within this article have been published before.
#########